# Early Detection of Slight Bruises in Yellow Peaches (*Amygdalus persica*) Using Multispectral Structured-Illumination Reflectance Imaging and an Improved Ostu Method

**DOI:** 10.3390/foods13233843

**Published:** 2024-11-28

**Authors:** Jian Wu, Chenlin Liu, Aiguo Ouyang, Bin Li, Nan Chen, Jing Wang, Yande Liu

**Affiliations:** 1Intelligent Mechanical and Electrical Equipment Innovation Research Institute, East China Jiaotong University, Nanchang 330013, China; wujian@ecjtu.edu.cn (J.W.); liucl_2617@163.com (C.L.);; 2National and Local Joint Engineering Research Center of Intelligent Photoelectric Detection Technology and Equipment for Fruit, Nanchang 330013, China; 3Institute of Quality Standard and Testing Technology for Agro-Products, Chinese Academy of Agriculture Sciences, Beijing 100081, China

**Keywords:** early detection, slight bruises, yellow peaches, multispectral structured-illumination reflectance imaging, I-Ostu method

## Abstract

Assessing the internal quality of fruits is crucial in food chemistry and quality control, and bruises on peaches can affect their edible value and storage life. However, the early detection of slight bruises in yellow peaches is a major challenge, as the symptoms of slight bruises are difficult to distinguish. Herein, this study aims to develop a more simple and efficient structured-illumination reflectance imaging system (SIRI) and algorithms for the early nondestructive detection of slight bruises in yellow peaches. Pattern images of samples were acquired at spatial frequencies of 0.05, 0.10, 0.15, and 0.20 cycle ${\mathrm{mm}}^{-1}$ and wavelengths of 700, 750, and 800 nm using a laboratory-built multispectral structured-illumination reflectance imaging system (M-SIRI), and the direct component (DC) and alternating component (AC) images were obtained by image demodulation. A spatial frequency of 0.10 cycle ${\mathrm{mm}}^{-1}$ and wavelength of 700 nm were determined to be optimal for acquiring pattern images based on the analysis of the pixel intensity curve of the AC image; then, the pattern images of all yellow peaches samples were obtained. The ratio image (RT) between the AC image and the DC image significantly enhances bruise features. An improved Otsu algorithm is proposed to improve the robustness and accuracy of the Otsu algorithm against dark spot noise in AC and RT images. As a comparison, the global thresholding method and the Otsu method were also applied to the segmentation of the bruised region in all samples. The results indicate that the I-Otsu algorithm has the best segmentation performance for RT images, with an overall detection accuracy of 96%. This study demonstrates that M-SIRI technology combined with the I-Otsu algorithms has considerable potential in non-destructive detection of early bruises in yellow peaches.

## 1. Introduction

The yellow peach (*Amygdalus persica*) is native to China and is named for the yellow color of its peel and flesh. Both fresh and canned yellow peach products are popular with consumers due to their rich nutrients, unique taste and flavor [1]. According to statistics, the production of yellow peaches in China is approximately 876,000 tons at present and maintaining a rapid rate of growth. However, during post-harvest processes such as picking, transportation, sorting, storage, and sale, yellow peaches are vulnerable to mechanical forces such as impact, vibration, and extrusion [2,3]. Mechanical damages are unavoidable throughout the post-harvest process due to the soft texture of yellow peaches, and bruising is the most common type [4]. The post-harvest loss of yellow peaches due to mechanical damages is as high as 20–30% [5]. Despite the commercialization of techniques for detecting fruit bruises, early detection of slight bruises in yellow peaches, which are similar in color to normal areas and are not easily detectable, remains challenging [6,7]. If slightly bruised fruits are not removed in time, especially during the hot and humid harvest period (usually summer), the bruised tissues deteriorate rapidly due to the production and action of ethylene gas and accelerate the softening of the surrounding healthy tissues [8]. This can even induce bacterial and fungal infection, leading to tissue decay and further infection of other uninjured fruits and resulting in significant economic losses and potential food safety problems [8]. Hence, it is essential to detect slight bruises in yellow peaches during the early stages to reduce the economic losses of producers and distributors.

With the rapid development of optoelectronic technology, the application of optical non-destructive detection techniques for detecting defects on fruit surfaces has increased significantly, such as machine vision (MV) [9], visible/near-infrared spectroscopy (Vis/NIR) [10], multispectral /hyperspectral imaging (MSI/HSI) [11,12], biospeckle imaging technique (BSI) [13], fluorescence imaging (FI) [14], thermal imaging(TI) [15], and X-ray imaging [16]. Although the techniques mentioned above have achieved certain success, they also have shortcomings in detecting fruit bruises. For instance, machine vision faces difficulty in identifying slightly bruised tissues with inconspicuous features [17]. Vis/NIR spectroscopy has a limited detection area and is prone to misjudgment [10]. The BSI technique needs to solve the issues of external noise and interference to obtain accurate results [13]. The application of the FI technique is limited due to the fact that many fruit varieties may not possess fluorescence characteristics [14]. Thermal imaging and X-ray imaging are not suitable for online detection of fruits at the commercial level [3]. The HSI technique is considered to be the most promising approach to study the detection of fruit qualities, and it can not only evaluate the internal quality of fruits (such as soluble solids, hardness, moisture, maturity), but also identify surface defects of fruits, because hyperspectral images can provide both spectral and spatial information [18,19,20]. However, HSI has limitations such as slow image acquisition speed and high equipment costs; as such, it is primarily used in laboratories and also not suitable for online detection or detecting subsurface defects.

The optical non-destructive detection techniques mentioned above are commonly used to identify surface defects and bruises by obtaining reflected images under uniform illumination (UI). However, it is difficult to realize the detection of bruised tissues below the fruit peel using those UI approaches due to the limited penetration depth of UI [21]. Recently, the field of agro-product non-destructive testing (NDT) has employed a new imaging technique, structured-illumination reflectance imaging (SIRI), based on structured light illumination (SI) [22]. Compared to UI, SI can control the depth of light penetration in tissues by changing the spatial frequency of illumination, enhancing the detection of defects on the surface or subsurface of fruits, and the maximum penetration depth of structured light illumination can amount to 1.2 mm. Therefore, SIRI has great potential for detecting defects that are difficult to distinguish visually on the subsurface of agro-products [23]. Based on the theory of light diffusion [24], the attenuation rate of structured light intensity in turbid substances is related to the spatial frequency. Hence, the penetration depth of the incident ray in the fruit tissue is determined by its spatial frequency, and with the increase in spatial frequency, the penetration depth decreases; on the contrary, the spatial resolution and contrast of the reflected image increase accordingly [23].

Prof. Lu Renfu’s research team from the USDA was the first to use SIRI to detect bruises in apples [25] and fresh pickling cucumbers [26]; they systematically investigated the potential of the SIRI system for bruising detection in agriculture products and developed a series of demodulation methods and detection algorithms to improve the detection speed and accuracy [27]. Due to the advantages presented by its wide field, depth discrimination, non-destructive detection, and low cost, SIRI technology has considerable potential for rapid non-destructive testing of hidden defects in agricultural products, and studies have recently been conducted on the detection of fruit defects such as the early decay of navel oranges [28,29,30], chilling injury of kiwi fruit [31], early damage in ‘Korla’ pears [32], and early decay of peaches [8]. Although previous studies have demonstrated the potential to detect multiple bruises in peaches using SIRI technology, the detection accuracy still needs to be improved, especially for early bruises [21,33]. Meanwhile, the current SIRI system only can acquire broadband grayscale images, which may be unfavorable for applications that require specific wavelengths. In general, a liquid crystal tunable filter (LCTF) is adopted for acquiring spectral images in specific wavelengths, which can quickly select any wavelength in the Vis/NIR range [12,29]. However, high-performance LCTFs are expensive and not suitable for the development of online detection equipment.

In addition, it is important to point out that the dark spot noise (uneven grayscale values) generated by the surface texture of yellow peaches during the three-phase image acquisition and demodulation process can easily lead to the incorrect recognition of normal yellow peaches. To achieve early detection of slight bruises in yellow peaches, this study proposes an easy to implement and low-cost multispectral structured illumination imaging system based on narrow-band filters. Then, the pattern images of samples at the optimal wavelength and spatial frequency were acquired for detecting bruises in yellow peaches, and the improved Otsu method was developed to improve the robustness of segmentation algorithms. In the hope of laying the foundation for the online and real-time detection of slight bruises in yellow peaches, the specific objectives of this study were the following: (1) Develop a narrow bandpass filter-based multispectral structured-illumination reflectance imaging (M-SIRI) system for acquiring spectral patterned images of yellow peaches. (2) Determine the optimal wavelength and spatial frequency for the detection of slight bruises in yellow peaches and acquire pattern images of all samples at the optimal parameters. (3) Develop and evaluate a segmentation algorithm for identifying yellow peaches with slight bruises in early stages using the improved Otsu, Otsu, and global thresholding algorithms.

## 2. Materials and Methods

### 2.1. Sample Preparation

On 20 August 2023, a total of 350 yellow peaches were purchased from an orchard located in Yanling County, Zhuzhou City, Hunan Province of China. After transportation to the laboratory, 310 yellow peaches with similar ripeness, uniform fruit diameters, and no obvious defects or contaminants on the surface were selected. All samples were randomly divided into two groups, a training set consisting of 50 normal samples and 100 slightly bruised samples, which were used to determine the appropriate algorithm and parameters. The testing set, consisting of 50 normal and 100 slightly bruised samples, was used to assess the performance of the algorithm, and the remaining 10 samples were used to determine the optimal wavelength and spatial frequency.

Generally, slight bruising is defined by pits impacted with a small ball with an appropriate impact energy hitting the surface of fruits, this is not easily recognizable when there is an intact and undamaged fruit peel. An impact test device was used to induce slight bruises on yellow peaches caused by mechanical external forces, as shown in Figure 1. The electromagnetic force of an electromagnet controls the fixation and release of a metal ball with a mass of 100 g and diameter of 30 mm. When the gravitational potential energy generated by the free fall of metal balls acts on yellow peaches, it can result in the plastic deformation and destruction of the shallow tissue of the flesh, leading to a certain degree of bruising. According to our previous study [20], an impact energy of 0.4 J induces slight bruises on the surface of yellow peaches that are not easily distinguishable. It is assumed that all the kinetic energy converted from potential energy of the metal ball is absorbed by the peaches. The impact energy is calculated by the following equation:(1)E=mgh
where *m* and *h* are the weight and the dropped height of the metal ball, respectively. The impact energy is 0.4 J with a fixed dropping height of about 400 mm.

Meanwhile, the falling ball impacts were randomly carried out in different areas of the yellow peaches such that the bruises were distributed in various locations to meet the requirements for the randomness of bruising in peaches. Since the pulp tissue will gradually deteriorate and change color under the action of enzymes and chemical reactions after bruising [4] resulting in samples that do not meet the requirements for early detection, the pattern images of the slightly bruised samples should be acquired within 2 h, and the room temperature and humidity should be the same before and after collection.

### 2.2. Multispectral SIRI System and Image Acquisition

The multispectral SIRI system (M-SIRI) based on the narrow-band filter was developed to obtain patterned images of yellow peach samples. As shown in Figure 2, the SIRI system mainly consists of a high-resolution (1920 × 1080 pixels) digital light projector (DLP6500, Texas Instruments, Dallas, TX, USA), a quartz tungsten halogen lamp (250 W, Ocean Insight, Shanghai, China), a grayscale camera (Baumer, Frauenfeld, Switzerland), a bandpass filter (T95-OD4-200-1100, Rayan Tech., Changchun, China), linear polarizer (GCM-0905M, Daheng Optics, Beijing, China), adjustable sample stage(Daheng Optics, Beijing, China), and a computer with an image acquisition card (mE5-MA-ACL, SiliconSoftware, Mannheim, Germany). The digital projector is to the side of the camera, which is located directly above the sample (perpendicular to the horizontal plane) and projects the incident light at an angle of about 15° to the vertical line. The spectral range of the light source covers the visible/near-infrared bands (360–2400 nm), the light output from the quartz tungsten halogen lamp transmits to the projector through an optical fiber, and the monochrome camera and the digital projector were connected to the computer through a data line with camera link interface and USB interface, respectively, to complete the projection of the sinusoidal pattern and the acquisition of the image. A pair of linear polarizers are mounted in front of the digital light projector and camera lens to suppress the specular reflection from the sample surface, and the narrow-band filter is installed on lens of the camera to acquire reflection images of the samples at a certain wavelength. To minimize the effect of ambient light on sample imaging, the M-SIRI system is operated in a closed dark-box, except for the computer and the halogen lamp.

When applying SIRI technology, the sinusoidal structured illuminations that are modulated at a certain spatial frequency are projected to the sample, and then the acquired three-phase shifting pattern images are demodulated into an alternating component (AC) image and a direct component (DC). The DC images correspond to uniform/diffuse illumination, while the AC images encodes depth resolution and contrast information, sub-surface or near-surface defects in fruits can be effectively enhanced [21]. Structured-illumination reflection imaging relies on the interaction between light and matter. According to the theory of light transmission, different substances have different absorption, scattering, and reflection characteristics of light. Healthy and bruised tissues differ in their sensitivity to different wavelengths of light. Due to the cloudy nature of fruit tissue, its function is similar to that of a low-pass filter, causing the energy of high-frequency light sources to rapidly decay within the tissues [24]. As shown in Figure 3, at the same depth of light transmission, the intensity of low-frequency light is greater than that of high-frequency light. Therefore, in the process of using SIRI technology to detect subsurface bruising in fruits, it is necessary to choose spatial frequency reasonably, allowing more photons to reach the depth of bruised tissues located in the subsurface, thus obtaining more information about bruises and providing benefits to the imaging of bruising features.

Based on the results of our pre-experiment and previous research [23], spatial frequencies of 0.05, 0.10, 0.15, and 0.20 cycle ${\mathrm{mm}}^{-1}$ are the most reasonable and are chosen for this study to evaluate the optimal spatial frequency for the detection of slight bruises in yellow peaches. Meanwhile, in order to determine the most suitable wavelength of structured illumination for the detection of slight bruises, the spectral images under certain wavelengths were acquired using narrow-band filters on the camera lens with center wavelengths of 700, 750, and 800 nm, respectively, since the AC images at wavelengths from 690 to 810 nm have a relatively strong image contrast and signal-to-noise ratio for bruise detection [27].

According to above analysis, the pattern images were acquired using spatial frequencies of 0.05, 0.10, 0.15, and 0.20 cycle ${\mathrm{mm}}^{-1}$ and wavelengths of 700, 750, 800 nm, with the samples placed on an adjustable sample stage with its stem–calyx axis in a horizontal orientation with the bruised area facing the camera. Three-phase sinusoidal structured illumination patterns with phase offsets of −2π/3, 0, and 2π/3 were generated using MATLAB 2022a (The MathWorks, Inc., Natick, MA, USA) with 8-bit grayscale images and projected onto the samples using the software supported by the digital projector. Subsequently, pattern images were synchronously acquired by a computer-controlled monochrome camera with an exposure time of 500 ms.

### 2.3. Image Preprocessing and Demodulation

The spatial intensity variations in the acquisition plane caused by halogen lamp and dark currents in the CCD camera can introduce excessive noise in the lower light intensity bands, which greatly affects the quality of the acquired pattern images. To reduce the effects of these factors, the original pattern images of each sample should be corrected with a white reference image and a black reference image, which are acquired by a lens on a quasi-white plate with a reflectivity of 98% and one covered with a lens cap, respectively. The calibration formula is shown in Equation (Equation 2).
(2)$$R=\frac{\left(R_{0}-B\right)}{W-B}$$
where *R*, $R_{0}$, *B*, and *W* are the corrected image, original image, black reference image, and white reference image, respectively.

Since the acquired pattern images with obvious streaks cannot be directly used to recognize the bruises in samples, it is necessary to demodulate the corrected three-phase pattern images to obtain the DC image and the AC image. Although the DC image is equivalent to a reflection image obtained under UI, the AC image contains depth information related to the spatial frequency of the sinusoidal illumination. The demodulation equations are as follows:(3)$$I_{AC}=\frac{\sqrt{2}}{3}\sqrt{{\left(I_{1}-I_{2}\right)}^{2}+{\left(I_{1}-I_{3}\right)}^{2}+{\left(I_{2}-I_{3}\right)}^{2}}$$
(4)$$I_{DC}=\frac{I_{1}+I_{2}+I_{3}}{3}$$
where $I_{1}$, $I_{2}$, and $I_{3}$ denote the pattern images acquired under sinusoidal illumination with phase shifts of −2π/3, 0, and 2π/3, respectively, and $I_{AC}$ and $I_{DC}$ denote the AC and DC images obtained using demodulation, respectively [34].

The demodulated images contain both yellow peach images and background information, and the background needs to be removed to facilitate the subsequent analysis of the yellow peach image, reducing the interference of the background information and enhancing the features of fruit bruises. Since the background intensity is usually lower than that of the yellow peach images, a suitable global threshold can be chosen to be applied to the demodulated image for background removal. A binarized mask was created using the DC images with a fixed threshold of 15; then, the mask is multiplied with the corresponding DC and AC images for background removal.

### 2.4. Image Enhancement

The original DC and AC images obtained by demodulation—usually with low grayscale values and low contrast between bruised and normal tissues—will greatly impact the effective identification of the bruise regions. Therefore, image enhancement is an important step in processing demodulated images for bruise detection. Firstly, automatic histogram equalization is performed on the original AC image, which results in significantly improved gray values.

The AC image emphasizes depth information but suffers from uneven illumination, manifested as uneven edge brightness and center brightness. The DC image reflects the overall reflectivity, which is equivalent to uniform illumination but contains only a small amount of information about the bruised area. In order to segment the bruising tissues while avoiding incorrect segmentation due to uneven brightness, The ratio image (RT) shows an effective image enhancement method by dividing the AC image with the DC image, thus improving the uneven illumination while retaining the information of the damaged region, thus enhancing the bruise features [28]. The RT image is calculated by the following equation:(5)$$RT=\frac{I_{AC}}{I_{DC}}$$
where $I_{AC}$ and $I_{DC}$ denote the AC and DC images obtained by demodulation, respectively.

### 2.5. Image Segmentation

Image segmentation is a critical procedure for the detection of bruised areas in yellow peaches after AC images enhancement. Thresholding is the most widely used technique for segmenting grayscale images. The Otsu algorithm is the most classic technique, known as the maximum interclass variance method. The principle of the method is to divide the image into two parts (background and foreground) according to the grayscale characteristics of the image [35]. The calculation process of the algorithm is as follows:

A digital image *I* of size *M* × *N* can be represented by *L* grayscale values [0, 1, 2, …, L−1]. The method is to find a threshold *T*∈ [0, *L*−1] dividing *I* into two classes $C_{0}$ and $C_{1}$. The mean and variance of $C_{0}$ and $C_{1}$ are denoted as $\mu_{0}$, $\mu_{1}$ and $\sigma_{0}^{2}$, $\sigma_{1}^{2}$, respectively. $p_{0}$ and $p_{1}$ denote the probability of classes $C_{0}$ and $C_{1}$, respectively. The threshold *T* is determined by searching the grayscale histogram of image *I* that minimizes the intraclass variance using the following:(6)$$arg\min_{T}p_{0}\left(T\right)\sigma_{0}^{2}\left(T\right)+p_{1}\left(T\right)\sigma_{1}^{2}\left(T\right)$$

When T is confirmed, the image *I* is separated into two classes $C_{0}$ and $C_{1}$. The sizes of $C_{0}$ and $C_{1}$ are $p_{0}MN$ and $p_{1}MN$. Equation (Equation 6) is equivalent to the maximization of between class variance, which is followed by Equation (Equation 7).
(7)$$arg\max_{T}\sigma_{W}^{2}=p_{0}\left(T\right)p_{1}\left(T\right){\left(\mu_{0}\left(T\right)-\mu_{1}\left(T\right)\right)}^{2}$$

The traditional Otsu method tends to yield over-segmentation, when there is dark spot interference in the image. Herein, a new objective function was designed based on the Otsu method, which is presented in Equation (Equation 8).
(8)$$\begin{array}{cc} arg\max_{T}\sigma_{W}^{2}\left(T\right)= & p_{0}\left(T\right)p_{1}\left(T\right){\left(\mu_{0}\left(T\right)-\mu_{1}\left(T\right)\right)}^{2}+{\left(\mu_{0}\left(T\right)-\mu \right)}^{2}+\\ & {\left(\mu_{1}\left(T\right)-\mu \right)}^{2} \end{array}$$
where μ is the average value of the whole image *I*, it can be expressed as Equation (Equation 9):(9)$$\mu =p_{0}\left(T\right)\mu_{0}\left(T\right)+p_{1}\left(T\right)\mu_{1}\left(T\right)$$

Unlike the standard Otsu method, the new objective function introduces two new terms aimed at measuring the distance between the mean of the image *I* and the mean of each segmentation class, balancing the weight of each class in determining segmentation thresholds, which can adjust the sensitivity to noise and help avoid over-segmentation [36].

As a comparison, the global thresholding method and the Otsu method were also applied to the segmentation of bruised regions in all samples. Firstly, a threshold is determined through experiments to ensure that it can remove most of the small scattered pixels, in order to reduce the interference of image noise. After removing connected regions with pixel values less than the threshold, if there are still non-zero pixels present, it is determined to be a bruised sample; otherwise, it is classified as normal. The best segmentation detection algorithm can be determined by calculating the accurate recognition rate of bruised and normal samples. Figure 4 shows a flowchart of the method to identify slightly bruised yellow peaches based on the M-SIRI system.

## 3. Results and Discussion

### 3.1. Image Demodulation and Background Removal

The sinusoidal structured illumination patterns projected onto the surface of the fruit as −2π/3, 0, and 2π/3 are phase shifted by the DLP6500 projector (Texas Instruments, Dallas, TX, USA). The pattern images acquired by the camera are demodulated to obtain DC and AC images, and the DC images have only a small amount of information about the bruised region, which is equivalent to the image acquired under a spatial frequency of 0 cycle ${\mathrm{mm}}^{-1}$ (uniform illumination), and the bruised region in the DC image has indistinct features. On the contrary, the AC images provide a great deal of information about the bruised areas, resulting in a strong visual contrast between the bruises and the normal areas, which suggests that M-SIRI can enhance the detection of slight bruises in yellow peaches. Since the DC images have better visual contrast between the fruit region and the background, they are more suitable for creating the mask for background removal. Mask creation was accomplished by applying a fixed threshold (T = 15) to the DC images of all samples. Then, background removal of the original DC (or AC) image can be achieved by multiplying the corresponding mask and applied to the AC and DC images of all samples for subsequent analysis.

### 3.2. Optimal Spatial Frequency and Wavelength Selection

Due to the fact that both spatial frequency and wavelength of illumination patterns affect the availability of acquired images for fruit bruise detection, it is critical to find an appropriate spatial frequency and wavelength for the detection of early bruises in yellow peaches. Figure 5 shows the AC images at the four spatial frequencies of 0.05, 0.10, 0.15, and 0.20 cycles ${\mathrm{mm}}^{-1}$ and the three wavelengths of 700, 750, and 800 nm. The corresponding pseudo-color images mapped using TWILIGHT. The brightness of the AC images could be reduced by increasing of the wavelength, which is accompanied by a decrease in the amount of light transmission. From the pseudo-color image, it can be seen that the image color difference of the 800 nm band is the smallest, indicating that the pixel intensity difference between the bruised area and the normal area is low, which is not conducive to the subsequent threshold segmentation. The image color difference of the 700 nm band is the largest, so the 700 nm band image is selected for subsequent image segmentation. The contrast of an AC image usually decreases with increasing wavelength due to the reduced photon efficiency of cameras and projectors as well as the lower reflectivity at longer wavelengths. This is why AC images at 800 nm have lower brightness and contrast.

At the same time, in one band, the image contrast changes with the change of spatial frequency, indicating that the bruise discrimination is directly related to the spatial frequency of sinusoidal structured illumination, but the surface features of yellow peaches (such as dents) also change with spatial frequency. Therefore, it is necessary to select a spatial frequency that is conducive to threshold segmentation. Figure 6, is an AC image at 700 nm, a corresponding pseudo-color image, and a pixel intensity curve. The pixel intensity curve corresponds to all pixels on the black solid line passing through the bruise area in the corresponding AC image. The peak-to-valley difference of the pixel intensity curve can reflect the contrast between the bruise area and the normal area in the AC image. As shown in the dotted area of Figure 6, the intensity variation on the sample surface is obvious, especially the pixel intensity difference between the healthy area and the bruise area. Therefore, the contrast between the normal area and the bruised area can be compared using the pixel intensity curve.

At the same spatial frequency, the AC images with a wavelength of 700 nm have the largest pixel intensity difference between the bruised and the normal regions, which is more conducive for bruise segmentation. Meanwhile, the images of 0.10 cycle ${\mathrm{mm}}^{-1}$ have the largest intensity difference, although the peaks of 0.05 cycle ${\mathrm{mm}}^{-1}$ are higher, but their intensity difference is smaller than that of 0.10 cycle ${\mathrm{mm}}^{-1}$. The peaks in the intensity distribution curves of 0.15 and 0.20 cycle ${\mathrm{mm}}^{-1}$ are lower than that of 0.10 cycle ${\mathrm{mm}}^{-1}$. Therefore, the wavelength of 700 nm and the spatial frequency of 0.10 cycle ${\mathrm{mm}}^{-1}$ were chosen as the optimal detection parameter in this study; then, the three-phase shift image of all yellow peaches samples were acquired with the optimal detection parameter.

### 3.3. Ratio Image and Image Enhancement

Compared to the DC and AC images, the RT images have better image contrast and more uniformed image intensity, which could increase the effectiveness of the segmentation algorithm in recognizing bruises. Figure 7A shows the RT image at a wavelength of 700 nm and a spatial frequency of 0.10 cycle ${\mathrm{mm}}^{-1}$ and the corresponding pseudo-color image. As can be seen from the pseudo-color image, the RT image has higher contrast and more uniform brightness than the AC image, and there are still some dark spots in the image due to the indentations on the surface of the yellow peach. Figure 7B shows the pixel intensity curves corresponding to the black solid lines in the RT image, the low pixel intensities of the edge regions were elevated to approximate the intensities of the center regions, the dark spots in the AC image was significantly improved in the RT image, and the difference in pixel intensities between the normal and bruised regions is more pronounced compared to the AC image. However, the image noise generated by the surface features of yellow peaches still existed in the RT images.

Figure 7C shows the pixel intensity distribution curves of the RT image after the median filtering of the 10 × 10 structural unit. The image noise was effectively eliminated by median filtering. Therefore, the RT image combined with median filtering can significantly reduce interference from dark spots in images and enhance the intensity contrast between the bruised and normal regions of yellow peaches.

### 3.4. Early Detection of Slight Bruise in Yellow Peaches

Figure 8 shows a flowchart for the detection of slightly bruised yellow peaches by M-SIRI, including three parts: (A) image acquisition and demodulation, (B) spatial frequency and wavelength selection, (C) segmentation and extraction.

Figure 9 shows the RGB, DC, AC, and RT images of five typical samples (four bruised yellow peaches and one healthy yellow peach). In the RGB image, the areas within the yellow circle represents the bruised area and the fifth sample is the normal yellow peach without any bruise. In the DC, AC, and RT images, the bruised areas were not obvious or even visible in the DC image but could be clearly observed in the AC image. However, the grayscale value of the AC image are lower and have more dark spot noise, which has a negative impact on subsequent bruising segmentation. The grayscale value of the RT image is more uniform, especially at the edge of the fruit, which is conducive to the accurate detection of yellow peaches with bruised areas at the edge. Overall, the characteristics of these three types of images are consistent with the previous analyses.

Figure 10 shows the segmentation images of yellow peaches obtained from five typical samples using three segmentation methods, including the Otsu, I-Otsu, and global thresholding methods. For illustration, the fruit contour was extracted and superimposed onto the defect segmentation result. The RT images have better effect than the AC images in all three segmentation methods, because the inhomogeneity of grayscale value in the AC images negatively affects the segmentation results, resulting in the absence of a valid bimodal histogram in the image to generate over-segmentation (note: a valid bimodal histogram is one peak representing the bruised tissue and the other peak represents sound tissue), and the DC images gave the worst performance due to the equivalent of uniform illumination. From the perspective of segmentation algorithms, the I-Otsu algorithm had the highest segmentation accuracy, with less mis-segmentation and over-segmentation, which indicated I-Otsu algorithm is more robust for images with uneven brightness levels. The algorithm with the next-highest segmentation accuracy is the I-Otsu algorithm, which also shows good segmentation results, but over-segmentation emerged in AC images due to dark spots with lower grayscale values. Moreover, the segmentation results of the global thresholding algorithm were the worst, with the occurrence of over-segmentation in the AC images and under-segmentation in RT images. In summary, the I-Ostu segmentation algorithm combined with RT images can achieve the most accurate segmentation.

Table 1 summarizes the results of the three segmentation methods for AC and RT images of all samples. The Otsu algorithm has the highest recognition accuracy for bruising detection in both AC and RT images, reaching 99.0% and 99.0%, respectively. The segmentation results of the global thresholding method are the worst, and the overall accuracies of AC and RT images are 90% and 89%, respectively. Compared to Otsu, the I-Otsu algorithm has the best overall segmentation accuracy of 96% for all 300 samples due to the better robustness for images with uneven grayscale regions. The results have demonstrated that the M-SIRI system combined with appropriate image enhancement and segmentation algorithms can provide superior capability of identifying yellow peaches with early bruising features. However, it is noted that normal samples have a high mis-classification rate in AC and RT images, with a total accuracy of <90%. This is mainly due to the interference caused by surface pits and the stem of the peach, which can be erroneously segmented and identified as bruises; this also highlights the limitations of threshold segmentation algorithms. Figure 11 sequentially displays the RT images and segmentation results of normal sample with surface pits, a sample with bruises located at the edge of the image, a sample with bruises located in the middle of the image, and an image with the stem and ventral suture. In Figure 11A, dark spot noise caused by surface pits leads to misclassification, where the Otsu algorithm experienced over segmentation. Figure 11D displays the segmentation results of the stem and ventral suture, where all three segmentation algorithms show mis-segmentation. The I-Otsu algorithm, by measuring the distance between the average value of the entire image and the average value of each segmentation class, can balance the weights of each class when searching for segmentation thresholds, thereby reducing the interference of image noise. It can effectively reduce misclassification caused by dark spot noise, stems, and ventral sutures. Further research can utilize the powerful feature extraction capabilities of deep learning network algorithms to improve the accuracy and efficiency of early bruises recognition.

The comparative analysis of methods for detecting bruises on yellow peaches is shown in Table 2. Li et al. [6] proposed an early bruise detection algorithm for yellow peaches based on hyperspectral imaging combined with an improved watershed segmentation algorithm, and the recognition accuracies of bruised peaches and intact peaches were 96.5% and 97.5%, respectively. Although hyperspectral imaging can be used for bruise detection in yellow peaches, the time-consuming acquisition and processing of hyperspectral limits its practical application. Sun et al. [21] based on structured hyperspectral imaging (S-HSI) to detect various types of post-harvest injuries in peaches, including impact, drop, and compression injuries that may lead to bruising, and applied artificial neural network and principal component analysis to establish a bruise detection model using S-HSI spectra with an average detection accuracy of 90.79%. Li et al. [20] utilized hyperspectral imaging in combination with a machine learning approach to differentiate between lightly bruised yellow peaches at different storage times and constructed an XGBoost model with an overall model accuracy of 95%, but the author concluded that the model had an insufficient sample size and could be risky during future external validation. Compared to the optical detection methods under uniform illumination, the M-SIRI technique obtained better accuracy in this study and also demonstrated a great deal of potential for commercial application due to its advantages of wide field, non-contact mechanics, and high speed. Moreover, the current SIRI system is only suitable for detecting the fruits in stationary situation, which significantly restricts its application. Faster image acquisition can be achieved by using two or even one pattern image demodulation method [30], and hardware should be improved by applying high speed camera and simultaneous projection and acquisition methods, so that the fruit movement during the acquisition of two consecutive phase-shifted images can be negligible [26]. Nevertheless, it is still a challenge to implement SIRI online real-time detection for fruits’ defects.

## 4. Conclusions

Sinusoidal pattern images of yellow peaches were acquired using a self-constructed M-SIRI system at wavelengths of 700, 750, and 800 nm and spatial frequencies of 0.05, 0.10, 0.15, and 0.20 cycles ${\mathrm{mm}}^{-1}$ and demodulated into the AC and DC images using the three-phase demodulation method. The wavelength of 700 nm and the spatial frequency of 0.10 cycles ${\mathrm{mm}}^{-1}$ produced the best contrast between bruised and normal areas by analyzing the pixel intensity curve in AC images. The AC images were more effective in resolving the bruised areas in the yellow peaches compared to the DC images, and the RT images after image preprocessing produced better image quality by reducing the brightness inhomogeneity. The image segmentation methods of I-Otsu, Otsu, and global thresholding were applied to the AC and RT images after image enhancement, the best segmentation results were achieved by using I-Otsu for the RT image with a total accuracy of 96%, improving the robustness of segmentation algorithms to images with uneven grayscale values. The result demonstrates the effectiveness of M-SIRI combined with the improved Otsu algorithm as a potential tool for early detection of slight bruises in yellow peaches, in the hope of providing reference for the application and development of SIRI online detection device for yellow peach bruising in the future. However, there are many effects that could impact the identification of slight bruises in actual fruits production, like surface pits, stems, and ventral sutures on peaches. Therefore, further studies should be conducted to evaluate how the presence of those properties affect the performance of M-SIRI in detecting slight bruises in yellow peaches. Moreover, the hardware and image analysis method also are need improved for implementing SIRI real-time inspection of yellow peaches for defects like bruising.

## Figures and Tables

**Figure 1 foods-13-03843-f001:**
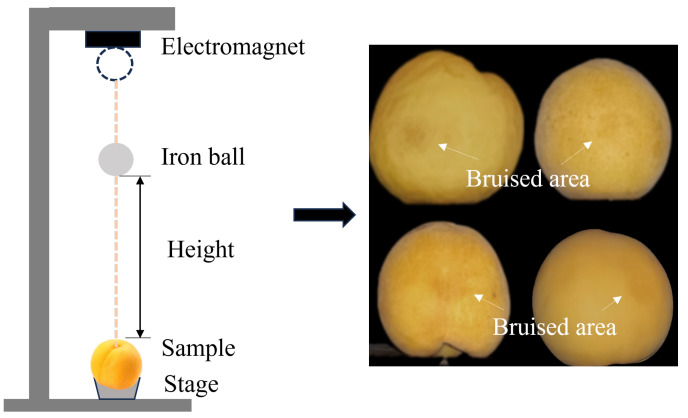
The impact test device and the slightly bruised yellow peaches after impact.

**Figure 2 foods-13-03843-f002:**
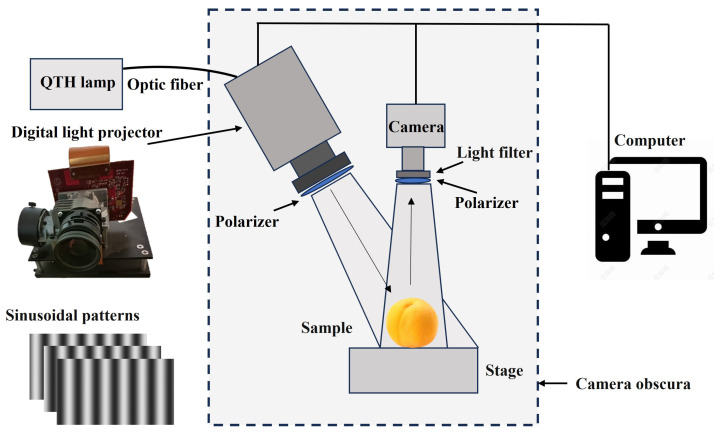
Schematic diagram of the multispectral structured-illumination reflectance imaging (M-SIRI) system.

**Figure 3 foods-13-03843-f003:**
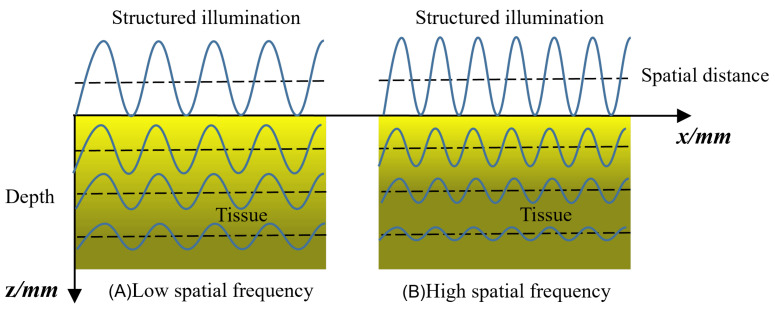
Schematic diagram of attenuation of structured light illumination in tissues. (**A**) Low spatial frequency. (**B**) High spatial frequency.

**Figure 4 foods-13-03843-f004:**
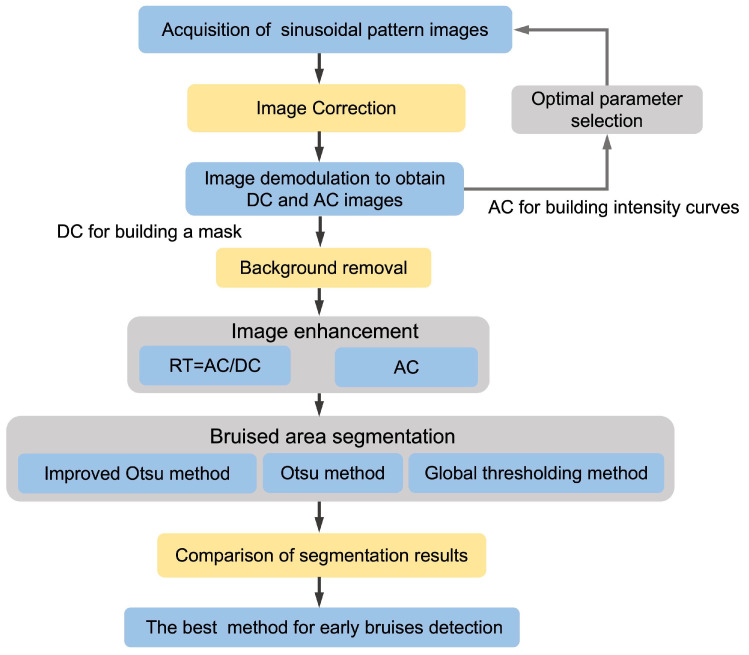
Flowchart of the M-SIRI technique for recognizing slight bruises in yellow peaches.

**Figure 5 foods-13-03843-f005:**
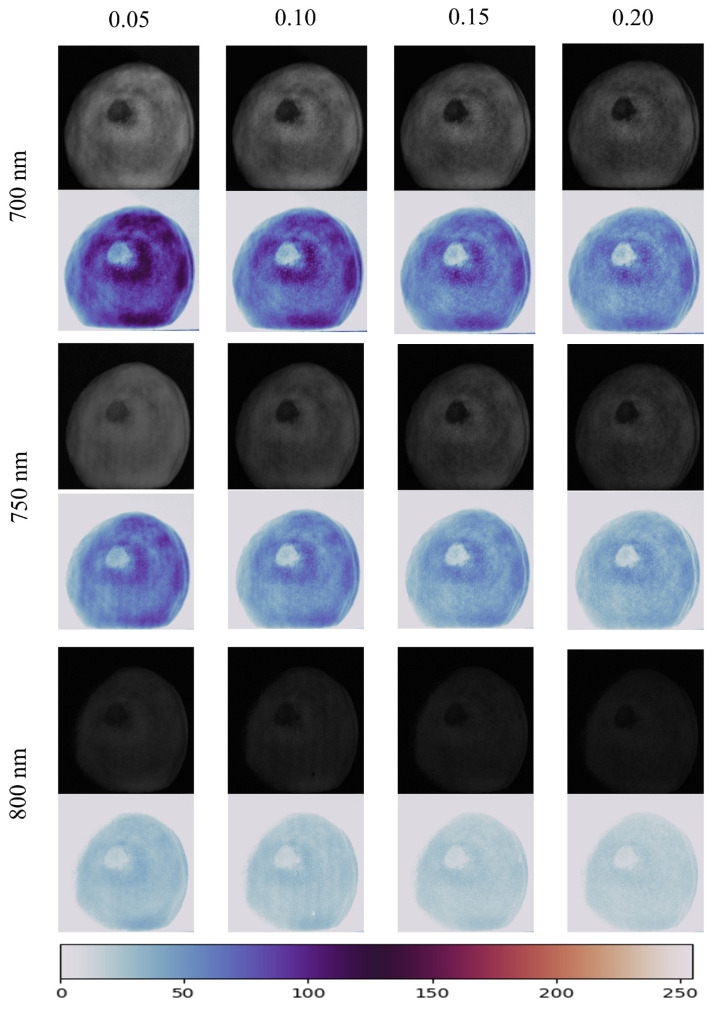
The alternating component (AC) images and corresponding pseudo-color images at four spatial frequencies (0.05, 0.10, 0.15, and 0.20 cycle mm^−1^) and three wavelengths (700, 750, and 800 nm).

**Figure 6 foods-13-03843-f006:**
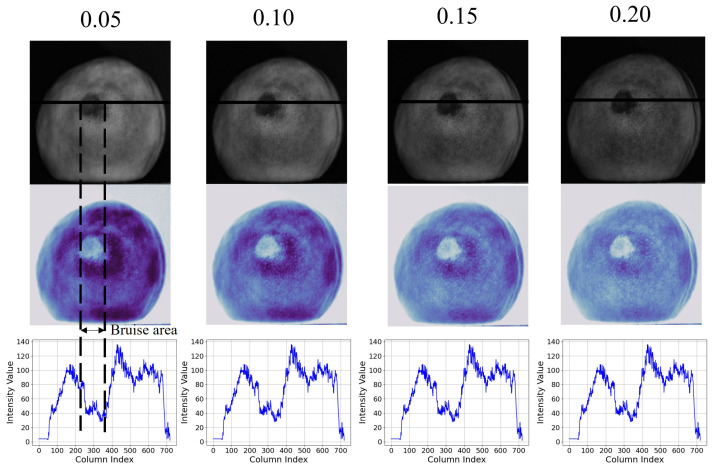
The alternating component (AC) images, corresponding pseudo-color images, and pixel intensity profiles for four spatial frequencies (0.05, 0.10, 0.15, and 0.20 cycles ${\mathrm{mm}}^{-1}$) in the 700 nm.

**Figure 7 foods-13-03843-f007:**
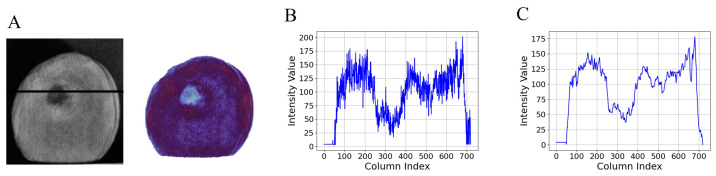
The ratio image (RT) image and intensity distribution curve. (**A**) The RT image with wavelength of 700 nm and spatial frequency of 0.10 cycle ${\mathrm{mm}}^{-1}$ and the corresponding pseudo-color image. (**B**) Intensity distribution curve before filtering. (**C**) Intensity distribution curve after filtering.

**Figure 8 foods-13-03843-f008:**
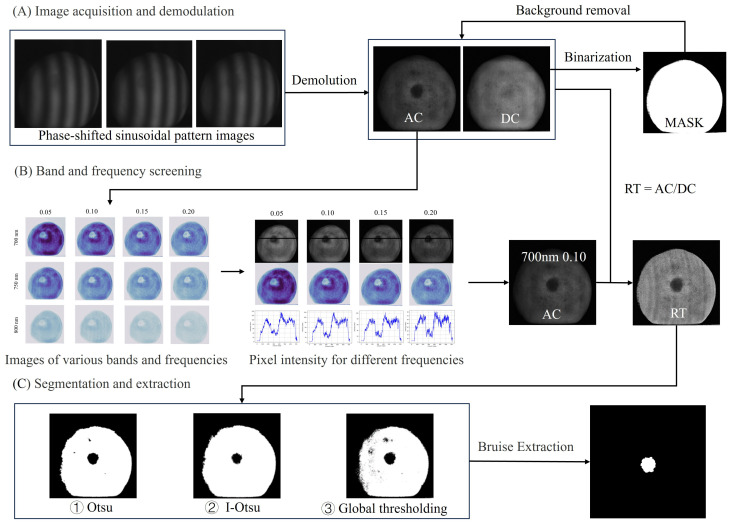
Flow chart for the detection of slightly bruised yellow peaches by M-SIRI. (**A**) Image demodulation and background removal. (**B**) Optimal spatial frequency and wavelength selection. (**C**) Segmentation and extraction of bruised areas.

**Figure 9 foods-13-03843-f009:**
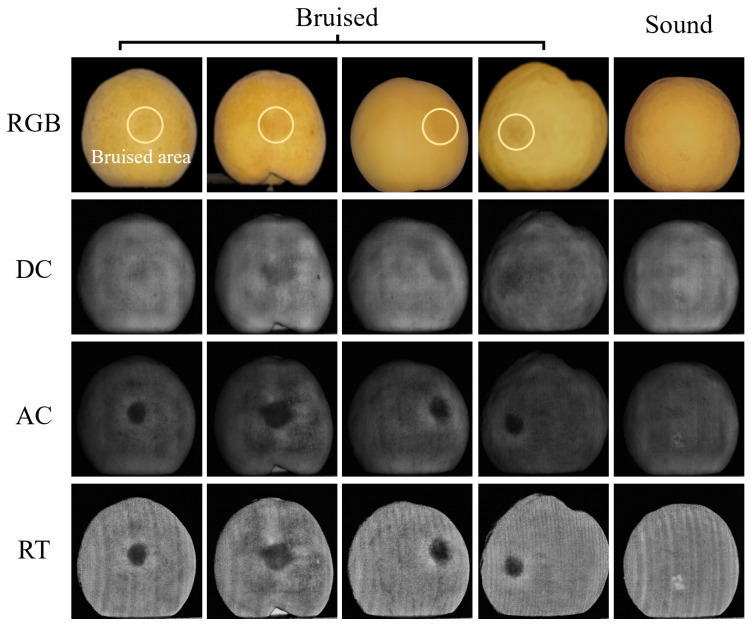
The RGB, DC, AC, and RT images of five typical fruit samples (The white circle indicates the bruise).

**Figure 10 foods-13-03843-f010:**
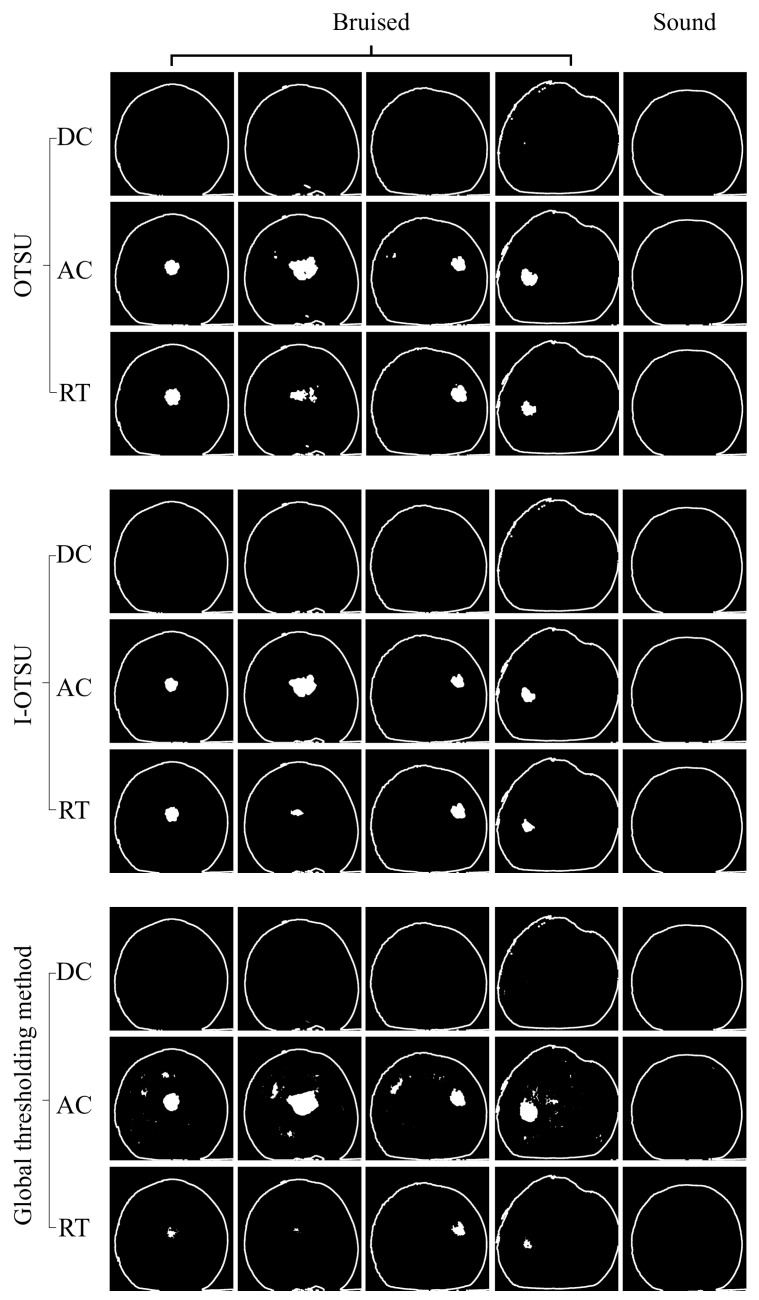
The segmentation results of Otsu, I-Otsu, and global thresholding algorithms for the DC, AC, and RT images.

**Figure 11 foods-13-03843-f011:**
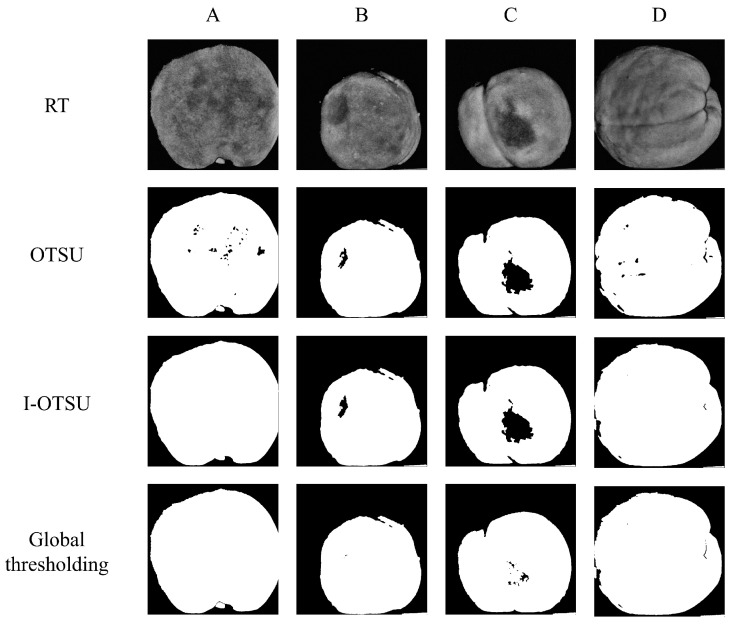
Mis-segmentations of several types of images in the three algorithms. (**A**) Normal sample with dark spot noise. (**B**) Bruising located at the edges of the image. (**C**) Bruising located in the middle of the image. (**D**) The image with the stem and ventral suture.

**Table 1 foods-13-03843-t001:** Classification accuracy obtained by using three segmentation algorithms for the AC and RT images.

Algorithms	Image Style	Sample Style	Training Set (n = 150)			Testing Set (n = 150)			Total (n = 300)
			**Normal**	**Bruised**	**Accuracy**	**Normal**	**Bruised**	**Accuracy**	**Accuracy**
Otsu	AC	Normal	45	5	90	42	8	84	87.0
		Bruised	1	99	99	1	99	99	99.0
									95.0
	RT	Normal	44	6	88	42	8	84	86.0
		Bruised	0	100	100	2	98	98	99.0
									94.7
I-Otsu	AC	Normal	43	7	86	42	8	84	85.0
		Bruised	2	98	98	2	98	98	98.0
									93.7
	RT	Normal	48	2	96	47	3	94	95.0
		Bruised	2	98	98	4	96	96	97.0
									96.3
Global thresholding	AC	Normal	38	12	76	35	15	70	73.0
		Bruised	1	99	99	2	98	98	98.5
									90.0
	RT	Normal	44	6	88	42	8	84	86.0
		Bruised	9	91	91	10	90	90	90.5
									89.0

**Table 2 foods-13-03843-t002:** The comparative analysis of methods for detecting bruises on yellow peaches.

Article	Technology	Algorithm	Accuracy	Detection Speed
Li et al. [6]	Hyperspectral	I-WSA	96.5%	Moderate
Sun et al. [21]	Structured Hyperspectral	ANN-DA	90.79%	Moderate
Li et al. [20]	Hyperspectral	XGBoost	95%	Slow
This Study	SIRI	I-Otsu	96%	Fast

## Data Availability

The original contributions presented in the study are included in the article, further inquiries can be directed to the corresponding author.

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
