# Peer review of "Early Detection of Slight Bruises in Yellow Peaches (Amygdalus persica) Using Multispectral Structured-Illumination Reflectance Imaging and an Improved Ostu Method"

_foods, 2024, doi:10.3390/foods13233843_

Round 1
Reviewer 1 Report
Comments and Suggestions for Authors
This manuscript aims to develop a simple and efficient structured illumination reflectance imaging system (SIRI) and algorithms for early nondestructive detection of slight bruises in yellow peaches. The system uses pattern images acquired at various frequencies and wavelengths. The optimal spatial frequency and wavelength are determined based on the pixel intensity curve of the AC image. The Otsu algorithm is improved to improve robustness and accuracy against dark spot noise in AC and RT images. The I-Otsu algorithm has the best segmentation performance for RT images, with an overall detection accuracy of 96%. These findings in the manuscript demonstrate the potential of The I-Otsu algorithm to provide new insights into early nondestructive detection of slight bruises in yellow peaches.
The paper is interesting. However, some Recommendations for Improvement should be addressed before publishing.
1. While the results show promise, they need to be presented more effectively with comprehensive comparative analysis against sate of art methods in a suitable table, especially text paragraph from lines 401 to 424.
Author Response
Comments 1: While the results show promise, they need to be presented more effectively with comprehensive comparative analysis against sate of art methods in a suitable table, especially text paragraph from lines 401 to 424.
Response 1: We appreciate the reviewers' positive feedback on our work and thank you for your review and valuable comments on our paper. We agree with your suggestion, so we have added a table to present these studies in a clearer way. This table compares the technology, algorithm, accuracy, and detection speed, as shown in Page 16 Table 2.
Reviewer 2 Report
Comments and Suggestions for Authors
Comments to Authors
1. Can you please update in detail why the early detection of slight bruises in yellow peaches is challenging, and why is it important for food quality control?
2. What are the advantages of using a structured-illumination reflectance imaging (SIRI) system for non-destructive bruise detection, in what other cases this system can be used?
3. What are the reasons due to which the spatial frequency of (0.10 cycle mm−1) and wavelength (700 nm) optimize the imaging process for detecting bruises?
4. What role does the ratio image (RT) between the AC and DC images play in enhancing bruise features, GLCM based texture analysis also signifies the texture features of the image.
5. How does the improved Otsu (I-Otsu) algorithm improve upon traditional segmentation methods for bruise detection accuracy?
6. What were the results of comparing the I-Otsu algorithm with other thresholding methods, and how does this impact future applications? Does this algorithm can be applied to other vegetable also.
7. What potential does M-SIRI technology, combined with the I-Otsu algorithm, produce for the broader applications in the non-destructive assessment of fruit quality?
8. If you can increase the legend size of the Figure 6 row 3, Figure 7b, this will great.
Reviewer 3 Report
Comments and Suggestions for Authors
The research examines the early identification of minor bruising in yellow peaches with a multispectral structured-illumination reflectance imaging (M-SIRI) method. The system acquires pattern images at several spatial frequencies and wavelengths, augmenting bruise characteristics via image demodulation. An enhanced Otsu algorithm is proposed to achieve superior segmentation accuracy. The research indicates that a wavelength of 700 nm and a spatial frequency of 0.10 cycle mm⁻¹ are ideal for identifying bruising. The M-SIRI system, in conjunction with the enhanced Otsu algorithm, attains a detection accuracy of 96%, indicating its efficacy for non-destructive early bruise identification in yellow peaches. The high detection accuracy of 96% is particularly noteworthy, as it demonstrates the potential for practical applications and shows a strong understanding of both the technical and practical aspects of the problem.
My main criticism is that the method lacks scale invariance, and the size of the bruise may have an impact on how it works. A description of the challenges may also be quite useful for effective system reproduction.
Otherwise, i really enjoyed the method and presentation. all the best
Author Response
Comments 1: My main criticism is that the method lacks scale invariance, and the size of the bruise may have an impact on how it works. A description of the challenges may also be quite useful for effective system reproduction.
Response 1: We thank the reviewers for their positive feedback on our work and for their valuable comments. We agree that bruise size can affect detection, and our preliminary experiments show that even smaller bruises show good accuracy, suggesting that scale variations may not be as influential as initially assumed. Please see the attached RT images and segmentation results. In this study, we aimed to develop and evaluate our algorithm under controlled conditions, and the effects of different injury levels are our next research goal. Therefore, bruises were created at a fixed height to maintain a consistent size. Although the current method has shown reliable performance, optimizing scale variance remains a future goal.